

# Transcriptome sequencing and analysis reveals the molecular response to selenium stimuli in *Pueraria lobata* (willd.) Ohwi

Kunyuan Guo[1], Yiwei Yao[2], Meng Yang[2], Yanni Li[2], Bin Wu[2] and Xianming Lin[1]

[1] Institute of Chinese Medicinal Materials, Hubei Academy of Agricultural Sciences, Enshi, China
[2] Chinese Academy of Medical Sciences and Peking Union Medical College, Institute of Medicinal Plant Development, beijing, China

## ABSTRACT

*Pueraria lobata* (willd.) Ohwi is a consumable selenium-enriched plant used for medicinal purposes. The molecular response to selenium (Se) stimuli in *P. lobata* is currently unknown. We used RNA-Seq to identify potential genes involved in selenite metabolism and analyzed their expression profiles. We obtained a total of 150,567 unigenes, of which 90,961 were annotated, including 16 structural genes, 14 sulfate transporters, and 13 phosphate transporters that may be involved in Se metabolism, and 33 candidate structural genes involved in isoflavone biosynthesis. The genes with a |foldchange| >2 and q value <0.05 after sodium selenite treatment were identified as differentially expressed genes (DEGs). We obtained a total of 4,246 DEGs, which were enriched in GO terms that included "response to stimulus", "response to stress", "signal transduction", "response to abiotic stimulus", and "response to chemical". Of the 4,246 DEGs, one sulfate transporter and five phosphate transporter genes involved Se metabolism, and nine structural genes involved in isoflavone biosynthesis were up-regulated. The expression patterns of 10 DEGs were selected randomly and validated using qRT-PCR. The Pearson Correlation Coefficient (r) was 0.86, indicating the reliability of RNA-Seq results. 22 Reactive Oxygen Species (ROS) scavenging DEGs were found, 11 of which were up-regulated. 436, 624 transcription factors (TFs) correlated with structural genes were identified that may be involved in Se and isoflavone biosynthesis, respectively, using r ($r > 0.7$ or $r < -0.7$). 556 TFs were related to at least one sulfate and phosphate transporter. Our results provided a comprehensive description of gene expression and regulation in response to Se stimuli in *P. lobata*.

## INTRODUCTION

*Pueraria lobata* (Wild.) Ohwi, known as kudzu, is mostly found throughout East Asian countries (*He et al., 2019*). The dried roots of *P. lobata* have been used medicinally for centuries to treat influenza, muscle stiffness, and other ailments (*The Chinese Pharmacopoeia, 2005*) with its usage first recorded in the *Shengnong Bencao Jing*, the

Corresponding authors
Bin Wu, bwu@implad.ac.cn
Xianming Lin, lxm31@sina.com

first comprehensive work on Chinese Medicine. Isolflavonoids, specifically puerarin and daidzein, are the main bioactive compounds found in the roots of *P. lobata* (*Du et al., 2010*). Modern pharmacological studies have revealed important functions of the *P. lobata* root. Puerarin is known to protect the cardiovascular system and may prevent osteoporosis, liver injury, and inflammation (*Wei, Chen & Xu, 2014*) and daidzein decreases blood alcohol levels (*Lowe et al., 2008*). A series of medicine, and health and food products have been developed using *P. lobata* as the functional ingredient because of its efficacy (http://db.pharmcube.com/database/cfda).

Selenium (Se) is a trace element essential for humans and animals and is an active ingredient of glutathione peroxidase, which is an antioxidant and has anti-aging and immunity benefits in humans (*Fairweather et al., 2011*). A severe selenium deficiency can lead to Keshan disease and Kashin-Beck disease while an overdose can cause hair loss, headache, fatigue and discoloration of the nails (*Fairweather et al., 2011*). Plants absorb selenite- and selenate-Se from the soil and transform them into organic Se (*Winkel et al., 2015*), which is the form predominantly consumed by humans. Current research has only speculated on the transport and transformational processes of selenate and selenite in higher plant species (*Winkel et al., 2015*). This process can be roughly classified into two stages: selenocysteine (SeCys) formation and Se methylation. In the SeCys formation stage, selenate may be taken up via sulfate transporters (*Terry, De Souza & Am, 2000*) and selenite may be taken up via phosphate transporters (*Barrow & Whelan, 2010*; *Terry, De Souza & Am, 2000*) or Si-transporters (*Zhao et al., 2010*). Selenite may be further reduced to selenide by sulfite reductase or by glutathione disulfide reductase and glutathione oxidoreductase. Selenide may then be transformed to selenocysteine (SeCys) by cysteine synthase, or by selenide water dikinase, Se-cysteine-tRNA synthase, and cysteine-tRNA ligase. SeCys may be incorporated into Se-proteins (*Aldwin Suryo & Davies, 2012*; *Hurst et al., 2013*), functioning as an antioxidant. SeCys may be transformed into dimethyl diselenide (DMDSe) or selenomethionine (SeMet) during the Se methylation stage and eventually into dimethyl selenide (DMSe) via the methionine cycle by a series of enzymes. DMDSe and DMSe can be emitted to the atmosphere to alleviate the Se toxification of the plant (*Winkel et al., 2015*).

Few studies have been performed on the molecular mechanism responsible for selenite stimuli in plants. One case study identified 14 selenite-responsive genes in *Astragalus racemosus* (*Hung et al., 2012*). Other research has shown that the antioxidative system was activated and photosynthesis and primary metabolism were enhanced under low selenite stress. High selenium stress has been shown to inhibit photosynthesis, primary metabolism, protein ubiquitination, and phosphorylation (*Wang, Wang & Wong, 2012*).

*P. lobata* is a selenium-enriched plant primarily composed of selenoproteins and selenopolysaccharides in which organic Se accounted for 82.42% of the total Se (*Du et al., 2010*). However, the molecular mechanism responsible for the Se stimuli response in *P. lobata* is unknown. RNA-seq was used to identify structural genes that may be involved in the biosynthesis of Se and isoflavonoids. We found that selenite may be taken up via phosphate transporters in *P. lobata*. Transcription factors (TFs) were found to

potentially regulate the structural genes in Se and isoflavonoid biosynthesis. Our results are foundational for engineering breeding cultivars of *P. lobata* with high Se compounds.

## MATERIALS AND METHODS

### Plant growth conditions and sodium selenite treatment

Mature stems were cut from one year-old *P. lobata* cv. GuiYege NO. 1 plants from Huazhong Medicinal Botanical Garden of the Institute of Chinese Medicinal Materials, Hubei Academy of Agricultural Sciences (Enshi, China). The stems were transplanted into sterilized vermiculite in plastic pots (16 cm ×16 cm) and grown in a greenhouse under a light/dark period of 16 h/8 h at 25 °C to control soil moisture and temperature. *P. lobata* seedlings with five leaves were treated with 200 mL sodium selenite at 0, 1, 5, 15, 25, 35, 55 mg/L, respectively, using an even spray on the vermiculite in the pots. The control was treated with distilled water. Every treatment was replicated three times.

### Measurement of physiological and biochemical indexes

The root length was measured on the 9th day after the sodium selenite treatment using a vernier caliper. Biochemical indexes were measured on the day of treatment and then then the 1st, 3th, 5th, 7th and 9th days of treatment thereafter. The total selenium content was measured using hydride atomic fluorescence spectrometry (*Fernanda et al., 2016*). The activity of superoxide dismutase (SOD) was determined as described by *Zhang et al. (2012b)* and the content of malondialdehyde (MDA) was assayed using the protocol from a previous study (*Feng et al., 2009b*). The correlation analyses of the differences between sampling days and concentrations of SOD and MDA were performed by SPSS 19 (*Qiu et al., 2017*).

### RNA extraction, library construction, and sequencing

Samples were collected on the day of treatment and the 1st, 3th, 5th, 7th and 9th days following the treatment with 25 mg/L sodium selenite. The roots were washed with distilled water, blotted with dry filter paper, and immediately frozen in liquid nitrogen. Total RNA was extracted, treated, and measured following the protocol from our previous study (*Wu et al., 2015*). Total RNA was extracted using the TRizol reagent according to the manufacturer's instructions and was digested to eliminate the residual genomic DNA using RNase-free DNase. The Agilent Technologies 2100 Bioanalyzer was used to measure and quantify the total RNA.

RNA-Seq libraries were constructed and sequenced as follows: the mRNA of each sample was separated from the total RNA using oligo (dT) magnetic beads; the mRNA were then cleaved into short fragments; the short fragments were used as templates and the first-strand cDNA was synthesized by reverse transcriptase and random primers. The RNA templates were then removed and the second-strand cDNA was synthesized using dNTPs, DNA polymerase I, and RNase H. These short, double-cDNA fragments were purified with DNA clean beads. The short cDNA fragments were ligated with the Illumina paired-end adaptors and purified with DNA clean beads following end reparation and A-tailing. PCR was used to selectively enrich DNA fragments with adapter molecules on both ends and

to create the final cDNA library. The quality of the cDNA library was measured using the Agilent 2100 Bioanalyzer. The libraries were then sequenced from the 5′ and 3′ ends using the Illumina sequencing system.

### *De novo* transcriptome assembly and annotation

RNA-Seq reads were *de novo* assembled using Trinity to obtain high-quality transcription sequences with the default parameters (*Grabherr et al., 2011*). The assembly unigenes were compared with sequences in the CDD database using BLAST (*Marchler et al., 2013*), KOG (*Koonin et al., 2004*), COG (*Tatusov et al., 2000*), NR and NT (https://blast.ncbi.nlm.nih.gov/Blast.cgi), PFAM (*Finn et al., 2016*), Swissprot (*Boeckmann et al., 2005*), TrEMBL (*Boeckmann et al., 2003*), GO (*Ashburner et al., 2000*) and KEGG (*Kanehisa et al., 2004*) for annotations with $E$-value $10^{-5}$ as the cutoff.

### Illumina data processing and gene expression profiles quantification

Raw reads were processed using in-house Perl scripts in FASTQ format. Clean reads were obtained after removing low-quality reads and those containing adapters and poly-N using Trimmomatic (*Bolger, Lohse & Usadel, 2014*). Gene expression levels were determined by TPM (Transcripts Per Kilobase of exon model per Million mapped read) to calculate the fold change. The read count was calculated using *Salmon* (*Patro et al., 2017*) and was loaded into DESeq2 (*Anders & Huber, 2010*) to calculate the $P$ value for differential expression analysis. Genes that had a |foldchange| $\geq 2$ and $q$ value $<0.05$ identified by DESeq were defined as differentially expressed genes (DEGs).

### TFs identification

Sequences of the unigenes were compared to the sequences in the iTAK database to analyze TFs in *P. lobata* using the default parameters (*Zheng et al., 2016*).

### Validation of differential expression genes by qRT-PCR

qRT-PCR were performed as described by our previous paper (*Wu et al., 2015*). 10 randomly selected genes were validated. Gene-specific primers were designed by Primer5 and the primer sequences are listed in Table S1. The 40S ribosomal protein S8 was used as an internal control (*He et al., 2019*). The Pearson Correlation Coefficient (r) was calculated between the mean results of qRT-PCR and RNA-Seq.

## RESULTS

### Physiological and biochemical index of *P. lobata* after sodium selenite treatment

The total Se content in *P. lobata* seedlings showed an increase and followed by a decrease with an increased sodium selenite concentration; plant leaves accumulated more Se than the stems. The total Se content in the seedling was lower when the concentration of sodium selenite was less than 5 mg/L, however the total Se content was maximized when the concentration was 25 mg/L. The accumulation of selenium in *P. lobata* seedlings began to decline when the concentration was more than 35 mg/L (Fig. S1). The root length tended to be the same under the Se accumulation and the sodium selenite treatments (Fig. S2).
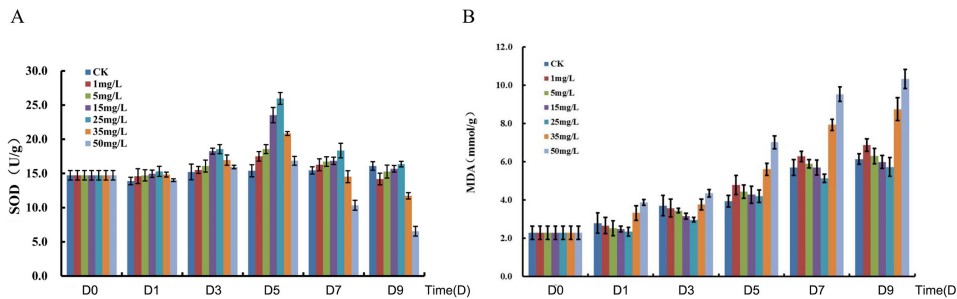

**Figure 1** **Measurement of SOD (A) and MDA (B).** D0, D1, D3, D5, D7 and D9 represent different day after sodium selenite treatment. Columns of different colors represent concentrations of sodium selenite.

SOD content showed an initial increase and then decrease. SOD reached its maximum content at 25 mg/L, 5 days after treatment, and then decreased gradually (Fig. 1A). The content of MDA tended to increase with the sodium selenite concentration and treatment time (Fig. 1B). Correlation analyses were conducted on the differences between sampling days and concentrations of SOD. MDA was also conducted. Results for SOD activity revealed that the treatment concentration of sodium selenite was negatively correlated on the 7th day and significantly negatively correlated on the 9th day (Table S2). The treatment from the 1st day was positively correlated with those of the 3rd, 5th, and 7th days. The treatment from the 3rd day was significantly positively correlated with that of the 5th day. The treatment from the 5th day was positively correlated with that of the 7th day. The treatment from the 7th day was significantly positively correlated with that of the 9th day. Results showed that the treatment concentration of sodium selenite was positively correlated the MDA content from the 1st day of treatment and was significantly positively correlated with the treatments from the 5th, 7th and 9th days, respectively (Table S3). There was a significant positive correlation between the treatment from the 1st day and those of the 3rd, 5th, 7th and 9th days, respectively.

## RNA-Seq sequencing and assembly

The 18 mRNA samples from days 0, 1, 3, 5, 7, 9 after sodium selenite treatment were sequenced on the Illumina platform. We obtained more than 6 Gb of data from every sample for the downstream analysis. The transcriptome data was assembled and 150,567 unigenes were obtained using Trinity (*Grabherr et al., 2011*). The N50 length and average read length of the assembly sequence were 1,742 bp and 962.46 bp, respectively. The length distribution and GC content of the unigenes was calculated (Fig. S3).

## Gene function annotation and categorization

The assembled 150,567 unigenes were compared with sequences found in the CDD, KOG, Nr, Nt, Pfam, Swissport, TrEMBL, GO, and KEGG by BLAST (version 2.2.26) databases using default parameters. There were 46, 439, 33, 401, 59,266, 69,142, 31,796, 58,101, 58,917, 64,796, and 5,415 resulting annotations found (Table S4). 90,961 total unigenes were annotated.
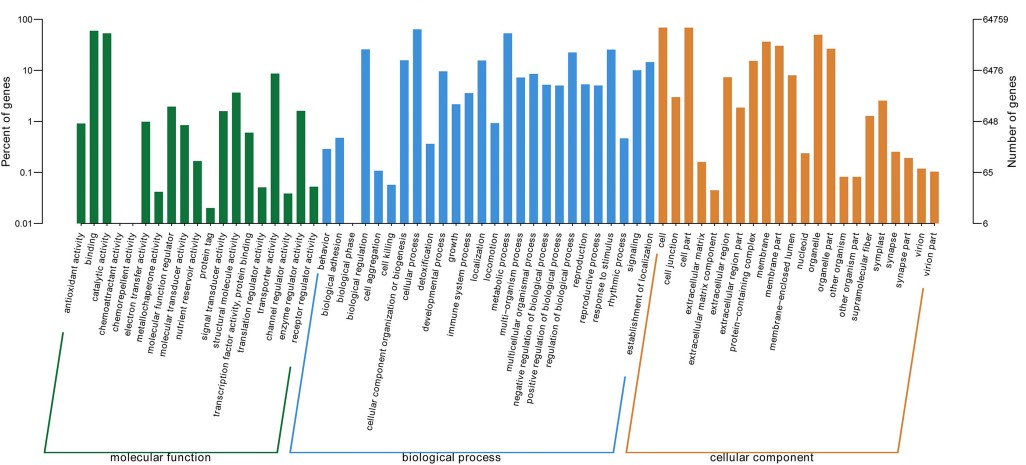

**Figure 2** **GO categories of *P. lobata* unigenes.** Percent of unigenes assigned to partial subcategories of the cellular component (blue), molecular function (red) and biological process (green) are represented.

GO ontology describes the molecular function, cellular component, and biological processes of genes. 67 total secondary classifications were obtained and the top five enriched GO terms were cell (GO: 0005623), cell part (GO: 0044464), cellular process (GO: 0009987), binding (GO: 0005488), and metabolic process (GO: 0008152) (Fig. 2).

## Identification of structural genes potentially involved in the biosynthesis of selenium and isoflavones

*P. lobata* is an important Se-enriched plant. However, its Se-related structure is unknown. The KEGG database was used to annotate and assign genes with Ko terms to specific metabolic pathways. 5,415 annotated genes in KEGG were further classified into four categories: genetic information processing, environmental information processing, metabolism, and cellular processes. The top five subcategories were signal transduction, carbohydrate metabolism, folding sorting and degradation, and overview (Fig. 3). A total of of 16 Se-related structural genes, 14 sulfate transporters, and 13 phosphate transporters were identified when KEGG was combined with Nr annotation (Table S5).

The isoflavone-related genes were further mined to determine whether the biosynthesis of the isoflavone in *P. lobata* was affected by sodium selenite treatment. The biosynthesis of isoflavones were divided into three stages: the phenylpropane pathway, the flavonoids pathway, and the isoflavone pathway (*Han et al., 2015a*; *Han et al., 2015b*; *He et al., 2011*; *Li et al., 2016a*; *Li et al., 2016b*; *Wang et al., 2016*; *Wang et al., 2017*). Five *PAL* s, three *C4H* s, and five *4CL* s were found in the phenylpropane pathway; three *CHS* s and three *CHI* s were found in the flavonoid pathway; one *IFS*, seven *HID* s, one *HI4OMT*, two *I7OMT* s, and three *IF7MAT* s were found in the isoflavone pathway. A total of 32 genes potentially involved in isoflavone biosynthesis in *P. lobata* were identified in this study (Fig. S4, Table S6).
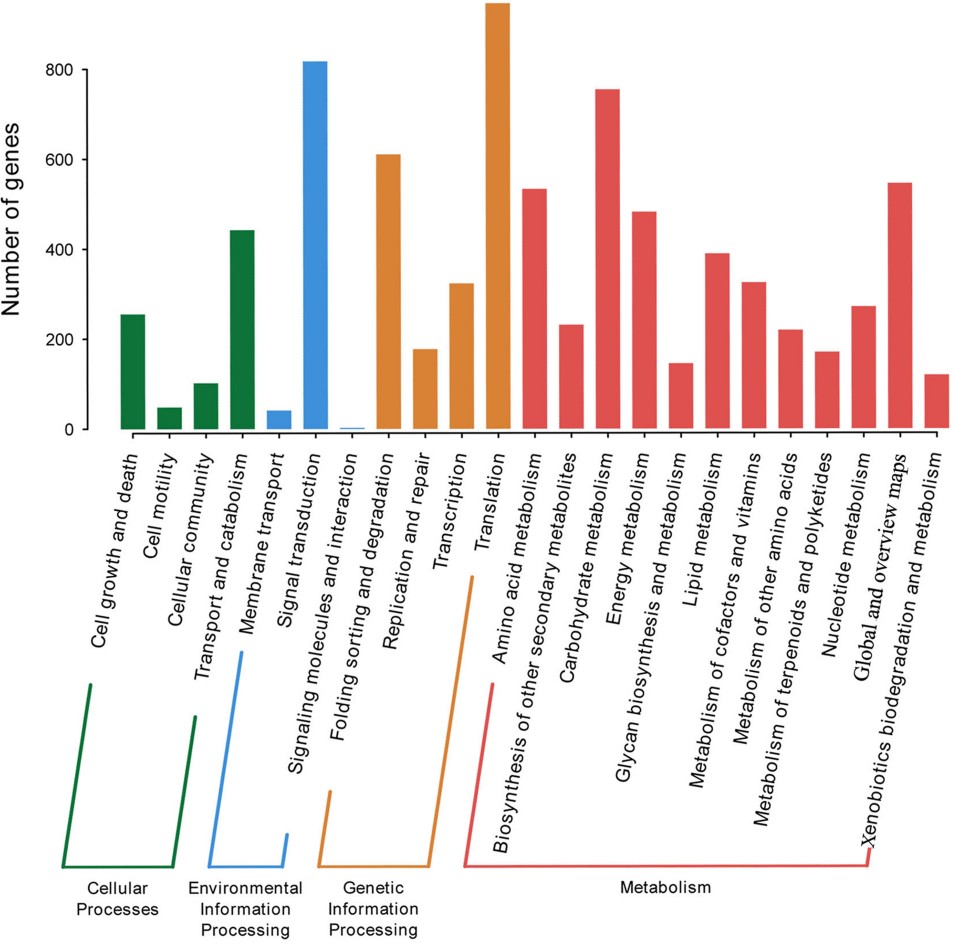

**Figure 3   KEGG categories of *P. lobata* unigenes.**

## Comprehensive analysis of DEGs

Genes with a |foldchange|>2 and *q* value <0.05 after sodium selenite treatments for different lengths of time were identified as differentially expressed genes (DEGs). A total of 4,246 DEGs were obtained (Table S7). 10 DEGs were selected randomly for analysis by qRT-PCR (Figs. 4A–4J) to validate the reliability of the RNA-Seq results (Table S8). The results between RNA-Seq and qRT-PCR were analyzed. The Pearson Correlation Coefficient (r) is approximately 0.86 (Fig. 4K), which indicates that the results of RNA-Seq and qRT-PCR are positively linearly related. The expression levels in RNA-Seq reflect the differences in gene expression after selenium treatment.

1,538 DEGs were identified after the first day of selenite treatment, including 465 up-regulated genes and 1,073 down-regulated genes. 1,386 DEGs were identified on the third day of treatment, including 559 up-regulated genes and 827 down-regulated genes. 1,348 DEGs were identified on the fifth day of treatment, including 407 up-regulated genes and 941 down-regulated genes. 2,510 DEGs were identified on the seventh day of treatment, including 883 up-regulated genes and 1,627 down-regulated genes. 1,556 DEGs

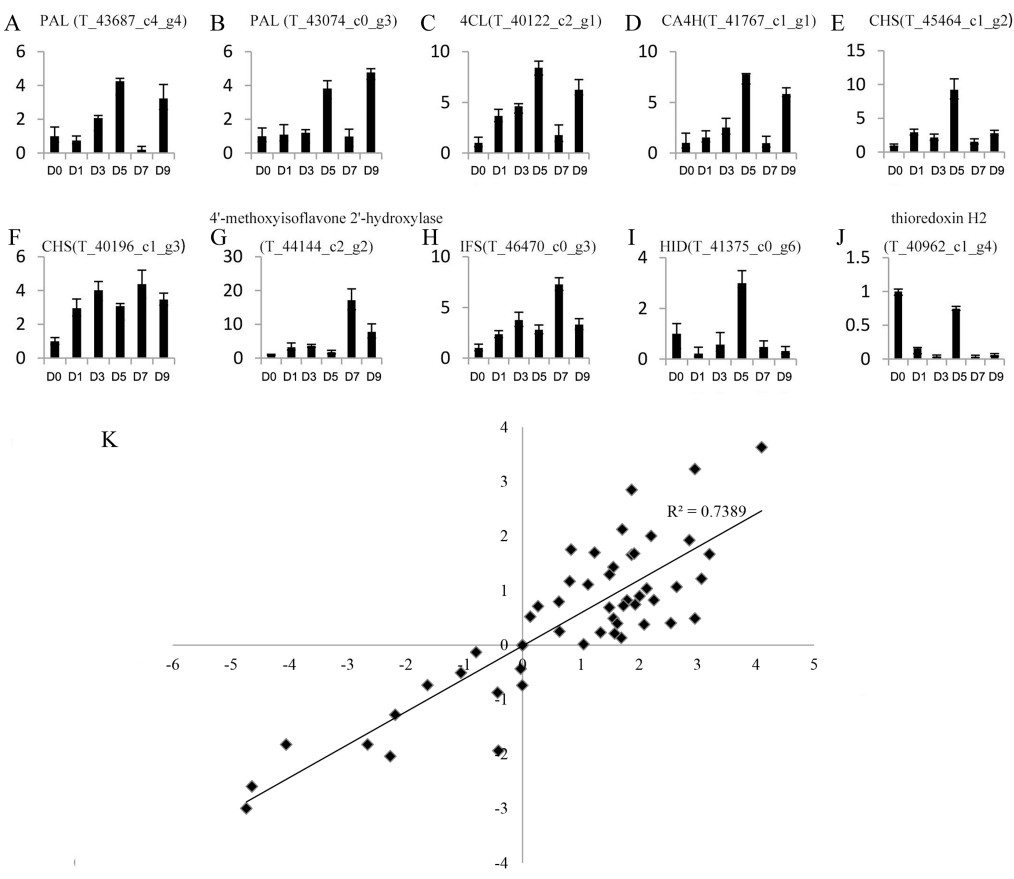

**Figure 4** **Expression profiles of 10 unigenes by qRT-PCR (A–J) and the correlation of the results between qRT-PCR and RNA-Seq (K).** (A–J) Fold changes of the unigenes are shown. The average expression levels in the controls were set to 1. Error bars represent standard error. (K) The correlation between the qRT-PCR and RNA-Seq expression levels. The $X$-axis represents log2 fold change in the expression levels found by qRT-PCR. The $Y$-axis indicates the log2 value of the expression level fold change from RNA-Seq.

**Table 1** **The number of up- and down-regulated DEGs based comparison with 0 day.** As control (|fold-change| >2; $q$ value <0.05).

| Comparison | Up-regulated | Down-regulated | Total |
|---|---|---|---|
| D1 vs D0 | 465 | 1073 | 1538 |
| D3 vs D0 | 559 | 827 | 1,386 |
| D5 vs D0 | 407 | 941 | 1,348 |
| D7 vs D0 | 883 | 1,627 | 2,510 |
| D9 vs D0 | 1,032 | 524 | 1,556 |

were identified on the ninth day of treatment, including 1,032 up-regulated genes and 524 down-regulated genes (Table 1).

We focused on the functional enrichment of the 4246 DEGs. GO enrichment showed that "response to stimulus", "response to stress", "signal transduction", "response to abiotic stimulus", and "response to chemical" were the top five classifications (Fig. 5), which

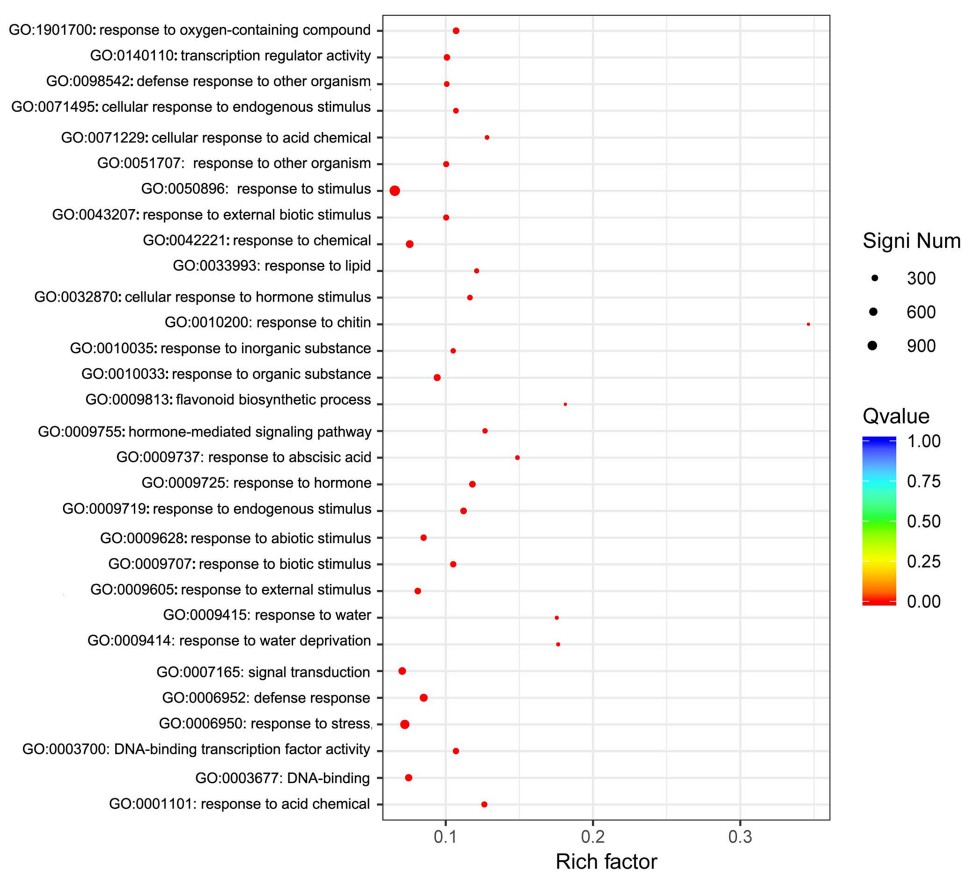

**Figure 5  GO enrichment of DEGs.** The $X$-axis represents the enrichment factor and $\log_{10}$ of the $Q$-value. The $Y$-axis indicates the different GO terms.

indicated that sodium selenite was a stress molecule. KEGG enrichment indicated that sodium selenite affected plant hormone signal transduction, MAPK signaling pathway-plant, starch and sucrose metabolism, ribosome biogenesis in eukaryotes, flavonoid biosynthesis, and isoflavonoid biosynthesis (Fig. S5).

Cluster analysis was performed on gene expression profiles of RNA-seq libraries from 18 samples according to Ernst's STEM algorithms arithmetic to assess the dynamic changes of DEGs after sodium selenite treatment (*Ernst & Bar-Joseph, 2006*). Four representative expression patterns were obtained (Fig. 6) and classified as subclusters. 1,443 genes were down-regulated as a whole after selenium treatment in subcluster 1; these were mainly involved in DNA metabolism, material transport, synthesis of primary metabolites, and stress-related processes. 881 genes in subcluster 2 were down-regulated on the first day and then up-regulated at another time, and were mainly involved in disease resistance, stress tolerance, material transport and cytoskeleton construction. 505 genes in subcluster 3 were up-regulated as a whole after selenium treatment; these were mainly involved in the secondary metabolism synthesis, response to toxin metabolism, and response to stress. 297

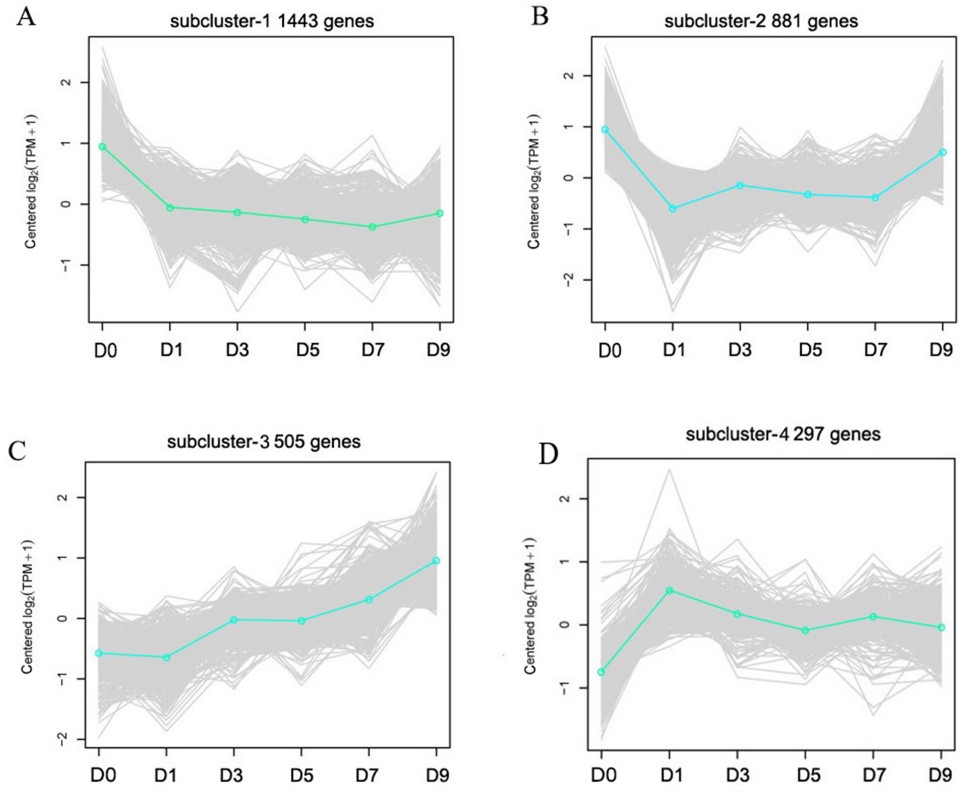

**Figure 6** **Representative time-course profile clusters.** (A–D) Four subclusters 1–4 are shown. D0, D1, D3, D5, D7 and D9 represent different day after sodium selenite treatment. For each time point and each gene, the centered log2 (TPM + 1) is indicted the mean expression profile for each cluster.

genes were up-regulated and then down-regulated on the first day in subcluster 4, and were mainly involved in regulating material transport, DNA repair, and resistance reactions.

## DEGs overlap at five different time points

DEGs were analyzed continuously after sodium selenite treatment. When compared with the control (Fig. 7), 160 DEGs were observed, including 101 up-regulated DEGs and 59 down-regulated DEGs (Table S9). Among the up-regulated DEGs, 9 genes (T_39191_c1_g1, T_39355_c1_g4, T_41331_c0_g6, T_42572_c0_g2, T_44141_c2_g4, T_44939_c1_g2, T_44939_c1_g3, T_44939_c1_g5, T_45070_c2_g1) are potentially involved in the biosynthesis of isoflavones; 1 gene (T_40246_c1_g1) encoded the selenium binding protein, one gene (T_42307_c2_g3) encoded glutathione s-type transferase and two genes (T_44116_c2_g1, T_44116_c2_g3) encoded phosphate transporters. Among those down-regulated genes, two MYB transcription factors, MYB39 (T_36389_c0_g1) and R2R3-MYB (T_41871_c2_g5), one thioredox protein (T_44132_c2_g1), and other genes that encoded TMV disease-resistant proteins and ion transport proteins were found.

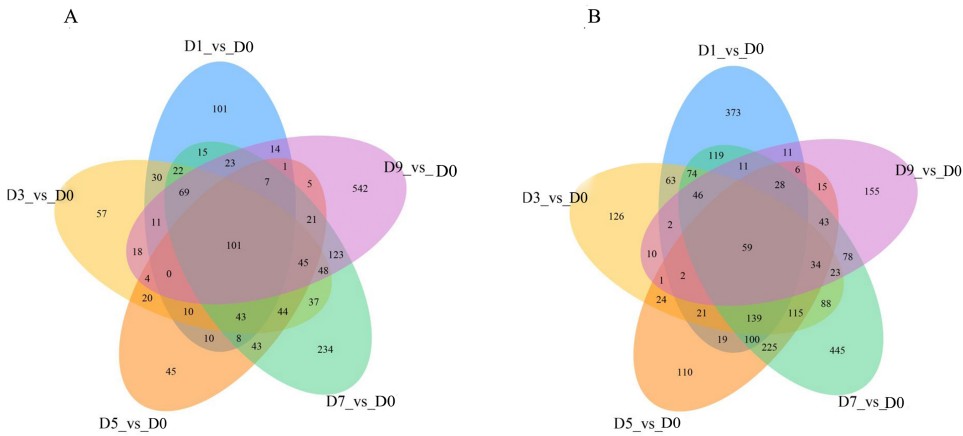

**Figure 7** The overlap of the DEGs are represented by Venn diagram (A) up-regulated genes and (B) down-regulated genes.

## Reactive Oxygen Species (ROS) scavenging genes in *P. lobata*

Low selenium stress has been shown to activate the antioxidant system in plants to scavenge ROS and enhance its ability to withstand a variety of oxidative stresses (*Wang, Wang & Wong, 2012*). ROS scavenging genes were mined from *P. lobata*, resulting in the up-regulation of 11 *SOD*s, 4 Catalase (*CAT*s), 4 Ascorbate peroxidase (*APX*s), 11 Glutathione peroxidase (*GPX*s), 55 Glutathione-S- Transferase (*GST*s), and 5 Monodehydroascorbate reductase (*MDHAR*s), in which 1 *SOD*, 1 *APX,* and 9 *GST*s were identified (Table S10).

## Transcription factor analysis

A total of 2,192 transcription factors (TFs) were identified by iTAK software, which were classified into 54 families (Table S11). Among the 2,192 TFs, 241 TFs representing 39 families were differentially expressed after selenium treatment. In order to clarify the regulatory role of TFs in selenium treatment in *P. lobata* co-expression analysis was performed with TFs among 75 genes, including 16 structural genes, 14 sulfate transporters, and 13 phosphate transporters genes related to selenium metabolism, and 32 structural genes potentially involved in isoflavone biosynthesis of *P. lobata*. 436 TFs were found to be related to at least one structural gene in selenium metabolism using the Pearson correlation coefficient r ($r > 0.7$ or $r < -0.7$) as the threshold (Table S12). 556 TFs were related to at least one of the sulfate transporters and phosphate transporters (Table S13). 624 TFs were related to at least one isoflavone-related gene (Table S14).

## DISCUSSION

*P. lobata* is a Se-enriched plant with an important role in Chinese herbal medicine (*Du et al., 2010*). However, the molecular mechanism responsible for its response to sodium selenite stimuli is unknown. In our study, the content of selenium in *P. lobata* under different concentrations of sodium selenite was investigated and revealed a pattern of an

increase followed by a decrease in selenium content. A maximum selenium content was achieved at a concentration of 25 mg/L of sodium selenite. The selenium content in leaves was higher than in the stems, which was consistent with results from other studies (*Han et al., 2013*; *Han et al., 2015a*; *Han et al., 2015b*). We measured the activity of SOD in *P. lobata* treated with different concentrations of sodium selenite. The results showed that an increase of treatment time did not significantly change the activity of SOD in the control group. However, an increased concentration of sodium selenite caused an increase in the activity of SOD in *P. lobata* followed by a decrease. The trend was most obvious on the fifth day of treatment when the concentration was 25mg/L. MDA content was determined and the results showed that MDA in *P. lobata* decreased and then increased with longer treatment times and greater concentrations. Our results indicate that the activity of SOD and the content of MDA in *P. lobata* could be increased using a treatment of low concentration sodium selenite. These results were consistent with results from previous studies (*Han et al., 2013*; (*Han et al., 2015a*; *Han et al., 2015b*). The root length, MDA content, and SOD activity under different concentrations of sodium selenite further supported 25 mg/L as a suitable treatment concentration.

The transciptome of *P. lobata* seedlings was sequenced after treatment with 25 mg/L sodium selenite and the DEGs were identified. 150,567 unigenes were obtained, of which 4,246 were DEGs. The enriched GO terms of DEGs respond to stimulus, stress, signal transduction, abiotic stimulus, and chemicals, which are similar to the responses of sodium selenate in *Astragalus chrysochlorus* (*Özgür et al., 2015*). This result is consistent with those of previous studies that the molecular mechanisms of plants that absorb selenite and selenate share some common pathways (*Li, McGrath & Zhao, 2008*; *Yu & Gu, 2008*). Sodium selenite may be taken up by phosphate transporters, because the expression profiles of 5 phosphate transporters genes were up-regulated after applying sodium selenite, and 2 genes (T_44116_c2_g1, T_44116_c2_g3) were continuously up-regulated. Phosphorus is known to be one of the macronutrients required for plant growth and development and a phosphate transporter is needed to absorb phosphate fertilizer. Therefore, we suggest that the absorption of sodium selenite and phosphate fertilizer may compete with one another. The use of these two fertilizers at the same time is contraindicated for the cultivation of Se-enriched plants.

ROS can damage cell membranes and other components and harm plants (*Nandini & Samir, 2016*). Low selenium stress has been reported to activate the antioxidant system to scavenge ROS and enhance a plant's ability to withstand a variety of oxidative stresses (*Wang, Wang & Wong, 2012*). We identified DEGs encoding major ROS scavenging enzymes, including SOD, CAT, APX, GPX, GST, and MDHAR (*Apel & Hirt, 2004*). We found that 1 *SOD*, 1 *APX*, and 9 *GST* s were up-regulated when compared with the control, which might enhance the anti-oxidation mechanisms after sodium selenite treatment in *P. lobata* (Table S7). GST (T_42307_c2_g3) was up-regulated consistently after selenite treatment.

To address the question of whether sodium selenite treatment in *P. lobate* would promote the biosynthesis of isoflavone, we analyzed the expression profiles of isoflavone-related structural genes and found that 9 genes were up-regulated (Table S6), indicating

that isoflavone biosynthesis was promoted after 25 mg/L sodium selenite treatment. These results are consistent with previous studies reporting that the total flavonoid and chlorogenic acid contents were enhanced when the application dose did not exceed 2.0 mg/kg (*Li, Sun & Liu, 2010*). These results suggest that the content of Se and other active compounds could be enhanced in medicinal plants by applying a suitable concentration of sodium selenite.

We identified the role of TFs in regulating selenium metabolism for the first time. 436 TFs were related with structural genes in Se metabolism (Table S12) and 556 TFs were co-expressed with sulfate transporters or phosphate transporters (Table S13). Our results provided a comprehensive overview of the gene expression and regulation in response to selenium stimuli in *P. lobata*.

## CONCLUSION

RNA-Seq was used to identify the molecular mechanism involved in the response to Se stimuli in *P. lobata*. We found 150,567 unigenes, of which 90,961 were annotated. Among these annotated genes, 16 structural genes, 14 sulfate transporters, and 13 phosphate transporters were potentially involved in Se metabolism and 32 structural genes were related to isoflavone biosynthesis. We obtained 4,246 DEGs, of which one sulfate transporter and five phosphate transporter genes involved in Se metabolism and nine structural genes involved in isoflavone biosynthesis were up-regulated. Furthermore, twenty-two ROS scavenging DEGs were identified, of which 11 were up-regulated. Finally, we identified some TFs that are potentially involved in Se metabolism and isoflavone biosynthesis. In order to cultivate Se-enriched plants, sodium selenite and phosphate fertilizer should not be applied at the same time. The content of Se and active compounds may be enhanced in plants used for medicinal purposes by applying a suitable concentration of sodium selenite.

### Funding

This study is supported by Special Project of Technological Innovation in Hubei province (2017AKB075) and the Science and Technology Program Research and Development Project of Enshi (D20180016). The funders had no role in study design, data collection and analysis, decision to publish, or preparation of the manuscript.

### Grant Disclosures

The following grant information was disclosed by the authors:
Special Project of Technological Innovation in Hubei Province: 2017AKB075.
Science and Technology Program Research and Development Project of Enshi: D20180016.

### Competing Interests

The authors declare there are no competing interests.

![PeerJ]

## Author Contributions

- Kunyuan Guo, Yiwei Yao, Meng Yang, Yanni Li, Bin Wu and Xianming Lin conceived and designed the experiments, performed the experiments, analyzed the data, prepared figures and/or tables, authored or reviewed drafts of the paper, and approved the final draft.

## Data Availability

The raw sequence data are available at the NCBI SRA database: PRJNA516771.

## Supplemental Information

Supplemental information for this article can be found online at http://dx.doi.org/10.7717/peerj.8768#supplemental-information.

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
