# Peer review of "Transcriptome sequencing and analysis reveals the molecular response to selenium stimuli in Pueraria lobata (willd.) Ohwi"

_PeerJ, doi:10.7717/peerj.8768_

## Round 0.1 · original submission · Major Revisions

Authors should prepare reasoned comments for both reviewers and make the necessary changes to the manuscript.

Reviewer 1 ·

Basic reporting

Except the introduction part, the English of the manuscript needs to be improved. In its current form, it is very difficult for the reader to understand the study. Please also check the formatting of the references and in-text citations, since it is not done in a consistent way (e.g., sometimes journal abbreviations are used, sometimes the full names; species names should be in italics etc.).
The introduction reads well, but in my opinion the paragraph about the medicinal effects of Pueraria (lines 41 to 52) needs some revisions. Several of the cited studies were conducted on rats. This is not clearly stated in the manuscript. The reader could get the expression that the mentioned medicinal effects are proven for humans.
Further, the order of the presented results should follow the order in the methods part.

Experimental design

The methods are well described (but see my comment on the English above), but some information is missing. Please indicate which software was used for read mapping (lines 112-113) and how much sodium selenite of different concentrations (line 119) was used for the treatment.

Validity of the findings

The analysis of SOD and MDA over the different sampling days is interesting (Fig. 1). I was wondering, whether the differences between sampling days as well as between concentrations are statistically significant. Maybe the authors could consider applying some statistical tests to provide more information about this.
In line 171 and Figure 3 the KEGG term “overview” is presented. Please check, whether this is correct. I have never heard about such a term.
In line 158 it is stated that 90961 genes were annotated. In the abstract, 43195 genes are reported to be annotated. Please check the correct number.
Please check the caption (and description in the Excel files) of Tables S9-S11. I think “corrected” needs to be replaced by “correlated”.
In the discussion part some more information could be added. In lines 252 to 254 it is stated that the investigation of MOD (I guess this is MDA) and SOD support 25mg/L as a suitable sodium selenite concentration. Here I miss some discussion about the content of SOD and MDA over the different treatments. For instance, can it be expected that SOD is decreasing, while MDA is increasing? Did other studies find similar results?
Furthermore, some sentences about the gene expression over the sampling days (Figure 5) could be added. I think this is an interesting result, which could be discussed.
In the conclusion part, the authors could also add some more information about the possible use of the results for other scientist and the cultivation of Pueraria lobata and could suggest some future studies.

Additional comments

The authors conducted a nice study, which revealed interesting results. The presentation of the results needs some improvement.

Reviewer 2 ·

Basic reporting

Guo et al. performed transcriptome sequencing to elucidate the response to selenium in Pueraria lobata (willd.) Ohwi. Here are some suggsetions:
Line 26. Person should be 'Pearson'
Line 46. there is extra tracking in sentence
Line 51. Rewrite the sentence starting 'All the studies...'
Line 54. rewrite 'our health' sentence
The second paragraph should be justified
Line 72, 103, 106 there is extra tracking in sentence
Line 127 '.' should be at the end of the lit cited
Line 129 omit Wu et al. in the parenthesis
Line 132 Person should be Pearson
Lİne 136 physilogical word starts with capital letter
Line 145 '.' should be at the end of the sentence
Sequencing results and physiological results should exchange places
Lİne 177, 215, 216, 217, 218 there is extra tracking in sentence
Lİne 227 The word 'indicates'
Validation of the expression of the gens could be given as last results
Line 265. there is extra tracking in sentence
In discussion 4th paragraph should be justified
discussion could be written with more details and could be developed

Experimental design

The research is within the aims and scopes of the journal.
Research question is well defined and meaningful.
Here are some suggestion:
In materials and Methods, I believe the method for SOD and MDA analysis should be described.
And also for RNA isolation and library construction for sequencing should be described in brief also.

Validity of the findings

I think the findings are meaningful and literatures are clearly stated. The data is robust and statistically sound.

However, I believe discussion could be given in more detail and could be developed. The weak part of the manuscript is discussion, I suggest to support the findings with other studies.

Additional comments

Guo et al. performed transcriptome sequencing to elucidate the response to selenium in Pueraria lobata (willd.) Ohwi. Here are some suggsetions:
Line 26. Person should be 'Pearson'
Line 46. there is extra tracking in sentence
Line 51. Rewrite the sentence starting 'All the studies...'
Line 54. rewrite 'our health' sentence
The second paragraph should be justified
Line 72, 103, 106 there is extra tracking in sentence
Line 127 '.' should be at the end of the lit cited
Line 129 omit Wu et al. in the parenthesis
Line 132 Person should be Pearson
Lİne 136 physilogical word starts with capital letter
Line 145 '.' should be at the end of the sentence
Sequencing results and physiological results should exchange places
Lİne 177, 215, 216, 217, 218 there is extra tracking in sentence
Lİne 227 The word 'indicates'
Validation of the expression of the gens could be given as last results
Line 265. there is extra tracking in sentence
In discussion 4th paragraph should be justified
discussion could be written with more details and could be developed
The research is within the aims and scopes of the journal.
Research question is well defined and meaningful.
Here are some suggestion:
In materials and Methods, I believe the method for SOD and MDA analysis should be described.
And also for RNA isolation and library construction for sequencing should be described in brief also.
I think the findings are meaningful and literatures are clearly stated. The data is robust and statistically sound.

However, I believe discussion could be given in more detail and could be developed. The weak part of the manuscript is discussion, I suggest to support the findings with other studies.

---

## Round 0.2 · Major Revisions

The manuscript needs professional editing, there are many grammatical errors.

At present, the title and conclusion is not supported by the findings; RNAseq is correlative and does not provide a mechanism.

RNAseq analysis is performed incorrectly; DEseq requires raw counts *NumCounts" column from Salmon, not TPM. Read the DEseq manual, 150,000 transcripts is too much; they need to be collapsed using a pipeline like DRAP http://www.sigenae.org/drap/.

Reviewer 1 ·

Basic reporting

see below

Experimental design

see below

Validity of the findings

see below

Additional comments

I thank the authors for addressing my suggestions. Before publication, the text should be carefully checked for typing errors such as was -> were (line 122), Swissport -> Swissprot (line 126), seedling -> seedlings (line 149), very -> every (line 172), or person -> Pearson (line 212).

---

## Round 0.3 · accepted · Accept

In the current form, the manuscript is ready for publication, the authors have substantially revised the text of the manuscript and improved it.